# A Method for Increasing the Robustness of Stable Feature Selection for Biomarker Discovery in Molecular Medicine Developed Using Serum Small Extracellular Vesicle Associated miRNAs and the Barrett’s Oesophagus Disease Spectrum

**DOI:** 10.3390/ijms24087068

**Published:** 2023-04-11

**Authors:** George C. Mayne, Richard J. Woodman, David I. Watson, Tim Bright, Susan Gan, Reginald V. Lord, Michael J. Bourke, Angelique Levert-Mignon, Isabell Bastian, Tanya Irvine, Ann Schloithe, Marian Martin, Lorraine Sheehan-Hennessy, Damian J. Hussey

**Affiliations:** 1Flinders Health and Medical Research Institute—Cancer Program, Flinders University, Bedford Park, SA 5042, Australia; george.mayne@flinders.edu.au (G.C.M.); richard.woodman@flinders.edu.au (R.J.W.); david.watson@flinders.edu.au (D.I.W.); tim.bright@sa.gov.au (T.B.); bell.bastian@hotmail.com (I.B.); ann.schloithe@flinders.edu.au (A.S.); marianmartin8296@gmail.com (M.M.); lorraine.sheehan-hennessy@sa.gov.au (L.S.-H.); 2Department of Surgery, Flinders Medical Centre, Bedford Park, SA 5042, Australia; susan.gan@sa.gov.au (S.G.); tanya.irvine@flinders.edu.au (T.I.); 3Gastroesophageal Cancer Research Program, St. Vincent’s Centre for Applied Medical Research, Darlinghurst, NSW 2010, Australia; rvlord@bigpond.com (R.V.L.); a.levert@amr.org.au (A.L.-M.); 4Faculty of Medicine and Health, The University of Sydney, Westmead, NSW 2145, Australia; michael@citywestgastro.com.au

**Keywords:** machine learning, cross validation, stability, robustness, bias–variance trade-off, biomarkers, microRNA, miRNA, Barrett’s oesophagus, oesophageal adenocarcinoma

## Abstract

The biomarker development field within molecular medicine remains limited by the methods that are available for building predictive models. We developed an efficient method for *conservatively* estimating confidence intervals for the cross validation-derived prediction errors of biomarker models. This new method was investigated for its ability to improve the capacity of our previously developed method, StaVarSel, for selecting stable biomarkers. Compared with the standard cross validation method, StaVarSel markedly improved the estimated generalisable predictive capacity of serum miRNA biomarkers for the detection of disease states that are at increased risk of progressing to oesophageal adenocarcinoma. The incorporation of our new method for *conservatively* estimating confidence intervals into StaVarSel resulted in the selection of less complex models with increased stability and improved or similar predictive capacities. The methods developed in this study have the potential to improve progress from biomarker discovery to biomarker driven translational research.

## 1. Introduction

Barrett’s oesophagus is estimated to occur in 1–2% of Western adults, and is characterised by formation of a metaplastic columnar cell epithelium in the distal oesophagus [1]. Patients with non-dysplastic Barrett’s oesophagus are at an increased risk (0.3–0.5% per year) of progressing through the stages of low-grade dysplasia and high-grade dysplasia to adenocarcinoma of the oesophagus [2]. The annual cancer risk is estimated at over 10% in patients with high-grade dysplasia [2,3]. Early detection of high-grade dysplasia or localised cancer has been reported to increase the 5-year survival rate to approximately 89% [4], compared with less than 20% for patients presenting with oesophageal adenocarcinoma [5].

We have previously demonstrated that circulating serum small extracellular vesicle (sEV) derived microRNAs (miRNAs) can be used as biomarkers to detect advanced oesophageal adenocarcinoma [6]. In the current study, we used miRNA sequencing to assess the potential of sEV derived miRNAs as early detection biomarkers. Sequencing was performed on serum sEV samples from patients without Barrett’s oesophagus (Controls), patients with non-dysplastic Barrett’s oesophagus (NDB; low cancer risk), and patients with high grade dysplasia (HGD; high cancer risk).

Biomarker based classification models derived from high-dimensional data, where there are typically many more candidate biomarkers than samples, are prone to overfitting. Various machine learning methods have consequently been developed to address this issue. The general approach with these methods is to determine an optimised level of constraint, within a cross validation framework, on the complexity of biomarker classification models. The aim is to optimise both the model-fit during development and the generalisability, known as the bias–variance trade-off. Development of an over-fitting model during training (small bias) will tend to result in a poor fitting model during external validation (large variance). This issue of poor generalisation also occurs in the historically standard approach to biomarker development involving only a single discovery cohort and an independent validation cohort [7]. However, a significant problem with the cross-validation approach is that it typically results in the selection of different biomarkers from each cross validation training set, which is often referred to as model instability [7].

The primary approach to reducing this instability in biomarker selection involves identifying the biomarkers that are selected frequently from the different cross validation training sets [8,9,10,11], and then determining the optimum number of these frequently selected biomarkers to include in the final model. We previously developed a stable variable selection method (StaVarSel) using this approach that selects a final stable biomarker prediction model in a way that maintains the required separation between the model selection process and the held-out test samples in a nested cross validation. This was achieved by the addition of an extra round of cross validations over the training sets within the nested cross validation framework [12].

However, when there is model instability, the variance of the prediction error also needs to be assessed in addition to the prediction error point estimate [11]. The standard way of estimating the variance of the classification error in cross validation has been to utilise a naive biased estimator [13]. The estimator is biased because the prediction errors are correlated across the training sets due to the overlap of the samples across the training sets. Although this is not an important issue when model selection is stable due to the inherently low level of variance, this naive approach can result in a significant underestimation of the variance when model selection is unstable [13].

We used leave-one-out cross validation (LOOCV) in the outer loop of StaVarSel to make stable biomarker selection computationally tractable. However, the use of LOOCV in the outer loop of a standard nested cross validation, which only produces a single prediction probability for each held out sample, has conceptually limited the range of methods available for estimating the variance of prediction errors to either the standard naive biased estimator that assumes that the prediction probabilities are normally distributed, or a non-parametric resampling method (e.g., bootstrap) that is applied over the prediction probabilities (one per sample) generated for the held out samples in the outer loop. Furthermore, no unbiased estimator of the variance exists for cross validation [13], and these standard methods can result in large underestimates of the variance (i.e., they can be anti-conservative) [14].

Various potential solutions to the problem of estimating the variance of cross validation prediction errors have been proposed [15]. However, these methods rely on assumptions or approximations that are not valid for the small sample sizes that are usually available for biomarker discovery studies [14,16]. It has therefore been suggested that the calculation of a *conservative* confidence interval that will not underestimate the variance, based on a holdout test, is the only rigorous and practically useful alternative for assessing classifier performance [16,17].

The development of clinically useful molecular biomarkers requires solutions to instability that are inherent in high throughput molecular data. StaVarSel is used within a nested cross validation framework to address this problem [12]. However, important issues remain regarding (i) how to estimate the classification capacity of training set derived biomarker models in a statistically *conservative* way so that the estimates are not optimistically biased and (ii) how to derive a *conservative* estimate of the generalisable prediction error for the final biomarker model. We therefore developed a method for *conservatively* assessing the variance of cross validation classification errors. We investigated the ability of this method to improve the robustness of the biomarker models selected by StaVarSel [12] and the potential utility of these methods for oesophageal disease biomarker development.

## 2. Results

### 2.1. Development of a Conservative Estimator of the Variance of Cross Validation Prediction Error

We had to use leave-one-out cross validation (LOOCV) in the outer loop of a nested cross validation in our previously implemented stable variable selection process (StaVarSel) to make the stable biomarker selection computationally tractable. However, the standard LOOCV framework, which only produces a single prediction probability for each held out sample, conceptually limits the range of methods that are available for estimating the generalisable predictive capacity of a set of stable biomarkers to those that can be overly optimistic (i.e., they are non-conservative).

In the current study, we determined that the estimation of the generalisable predictive capacity of a set of stable biomarkers does not need to be restricted to the standard LOOCV framework. Each LOOCV training set can be resampled repeatedly to produce a range of prediction probabilities for each corresponding held out sample from which *conservative* non-parametric 95% confidence intervals can be derived ([18]; see Methods Section 4.10 for a detailed explanation). The bounds of the confidence intervals for all samples can then be used simultaneously, and therefore also *conservatively* [19], to derive worst case estimates of specificity and sensitivity from the misclassification rates for each of the health states under consideration.

To investigate the properties of this *conservative* prediction error variance estimator, we compared it with a non-conservative bootstrap resampling method and with the naive biased method that assumes that the prediction probabilities are normally distributed. We applied these methods to serum sEV miRNA data, from samples relevant to the early detection of oesophageal adenocarcinoma, in a standard nested cross validation which used miRNA-ratios selected by Lasso regression to build a generalised linear logistic regression model within each training set (i.e., a relaxed Lasso; refer to Section 4.9 for more information). The results from applying this resampling based *conservative* estimator are presented in Figure 1b,d (prediction probability plots), and for the non-conservative bootstrap estimator in Figure 1a,c (ROC curves). The specificity and sensitivity values and their 95% confidence interval lower bounds are also presented for each plot in Figure 1 to enable quantitative comparisons. The non-conservative bootstrap method produced slightly higher estimates compared to the naive biased estimator for the confidence interval lower bounds on the accuracy ((specificity + sensitivity)/2) for Controls vs. NDB (43.1% vs. 35.9%) and NDB vs. HGD (35.6% vs. 29.6%). Our *conservative* estimator produced considerably lower estimates than either of these non-conservative methods for the lower bounds on the accuracy for both Controls vs. NDB (6.3%) and NDB vs. HGD (7.7%) (Figure 1 and Appendix A).

### 2.2. Comparison of Stabilised (StaVarSel) vs. Standard Nested Cross Validation

We initially investigated whether increased levels of regularisation would improve the classification models but this resulted in no meaningful gains in predictive capacity (Appendix A; see Methods Section 4.8 for details). These results are consistent with our previous observations with serum miRNAs from patients with oropharyngeal squamous cell carcinoma [12]. We subsequently investigated the capacity of our StaVarSel stable variable selection method to produce useful models for our oesophageal disease pathway miRNA data in comparison with standard nested cross validation. StaVarSel produced models with increased and potentially clinically useful predictive capacities relative to the standard nested cross validations for both Controls vs. NDB (97.6% vs. 64.1% accuracy; Figure 2c vs. Figure 1a;) and for NDB vs. HGD (97.6% vs. 56.9% accuracy; Figure 3c vs. Figure 1c).

### 2.3. The Conservative Prediction Error Variance Estimator Identified Less Complex Models

We subsequently assessed the effects on model complexity and on the estimates of generalisable prediction accuracy of our *conservative* method for estimating the variance of prediction errors within StaVarSel stabilised nested cross validation. For Controls vs. NDB, the non-conservative bootstrap resampling method for estimating the variance resulted in an optimal model containing 6 miRNA-ratios (Figure 2c), whereas our *conservative* sub-sampling per sample method resulted in a less complex model containing 2 miRNA-ratios (Figure 2b). For NDB vs. HGD, the non-conservative bootstrap resampling method resulted in an optimal model containing 7 miRNA-ratios (Figure 3c), whereas the *conservative* sub-sampling method resulted in a less complex model containing 4 miRNA-ratios (Figure 3b).

These biomarker models were assessed, using both the non-conservative and the *conservative* variance estimators, for their generalisable classification accuracy using the held-out samples in the outer loop of the nested cross validation. For Controls vs. NDB, the *conservatively* derived optimum 2 miRNA-ratio model (Figure 2b) produced higher *conservative* estimates of specificity (84.6%) and sensitivity (90.5%) at the confidence interval lower bounds than both the *conservative* estimates for the non-conservatively derived optimum 6 miRNA-ratio model model (76.9% and 71.4%; Figure 2d) and the non-conservative estimates for the *conservatively* derived optimum 2 miRNA-ratio model (61.5% and 85.7%; Figure 2a). The non-conservative estimates of the specificity (100%) and sensitivity (85.7%) for the non-conservatively derived optimum model (Figure 2c) were optimistic relative to the *conservative* estimates (Figure 2d).

The *conservatively* derived optimum model for Controls vs. NDB (Figure 2b) contained the following miRNA ratios: miR-181b-5p/miR-328-3p and miR-21-5p/mir-126-3p, and the non-conservatively derived optimum model (Figure 2c) additionally selected miR-501-3p/miR-328-3p, miR-106b-3p/miR-103a-3p, miR-21-5p/miR-103a-3p, and miR-99a-5p/miR-30a-5p. These miRNA ratios were selected from the most frequently selected miRNA ratios via an additional inner layer of cross validation (see Section 4.9 and Supplementary schema for details). The levels of each of these six individual miRNA-ratios are shown in Appendix A, and details of the miRNAs in these ratios are in Appendix A.

For NDB vs. HGD, the *conservatively* derived optimum 4 miRNA-ratio model (Figure 3b) produced *conservative* estimates at the confidence interval lower bounds for specificity (81.0%) and sensitivity (76.5%) that were similar to the *conservative* estimates for the non-conservatively derived optimum 7 miRNA-ratio model (76.2% and 82.4%; Figure 3d), and higher than the non-conservative estimates for the *conservatively* derived 4 miRNA-ratio model (76.2 and 70.6%; Figure 3a). As with Controls vs. NDB, the non-conservative estimates of the specificity (85.7%) and sensitivity (100%) for the non-conservatively derived optimum NDB vs. HGD model (Figure 3c) were optimistic relative to the *conservative* estimates (Figure 3d).

The conservatively derived optimum classification model for NDB vs. HGD (Figure 3b) contained miR-324-5p/let-7b-5p, miR-17-5p/miR-126-5p, miR-146a-5p/miR-361-5p, and miR-126-5p/miR-152-3p. The non-conservatively derived optimum model (Figure 3c) additionally selected let-7i-5p/miR-126-5p, let-7b-5p/miR-151a-3p, and miR-423-5p/miR-483-3p. The levels of each of these seven individual miRNA-ratios are shown in Appendix A, and details of the miRNAs in these ratios are in Appendix A.

The *conservative* estimates for the *conservatively* derived models (Controls vs. NDB, 2 miRNA-ratio model; NDB vs. HGD, 4 miRNA-ratio model) produced the smallest 95% confidence intervals overall for specificity and sensitivity for the held out samples (Table 1), which indicated that the *conservatively* derived models are likely to be more robust than the non-conservatively derived models when applied to new samples.

## 3. Discussion

The derivation of robust, reproducible molecular biomarkers from high dimensional data is challenging and requires the application of machine learning combined with unique statistical methods to solve issues with classification model instability and with the estimation of the variance of classification errors.

Various methods have been proposed for stabilising biomarker selection within a cross validation framework. However, a significant challenge for stable biomarker selection has been maintaining the separation between the model selection process and the samples that are held out for estimating the generalisable predictive capacity of the final model [20,21]. We previously developed StaVarSel, an implementation of stable variable selection that addresses this problem by using an additional round of cross validations across the training sets within a nested cross validation to select an optimum number of stable biomarkers [12].

If model selection is affected by instability, then an estimate of the variance in the classification error is required for determining the optimum model complexity. We previously utilised a non-parametric bootstrap estimator of variance within StaVarSel [12]. However, no unbiased estimator of variance exists for cross validation [13], and it has been suggested that *conservative* estimates that do not underestimate the variance could be derived using holdout tests [14,22]. Therefore, the main goal of our current study was to develop a *conservative* non-parametric estimator of the variance of cross validation prediction errors using hold out tests. While this is more computationally intensive than non-conservative bootstrap resampling of the LOOCV prediction probabilities, and can take several hours to run, it is tractable for moderate sample sizes (e.g., up to 50) on a standard desktop or laptop computer. It should also be noted, however, that this approach needs to be used and interpreted with caution because it has the potential to produce overly conservative confidence intervals [22].

We applied our solutions for both reducing cross validation instability and for *conservatively* estimating prediction error variance to our circulating serum sEV derived miRNA sequencing data. Our stable variable selection method, combined with our *conservative* prediction error variance estimator, markedly improved the predictive capacity of the biomarker models compared with standard nested cross validation, and reduced the complexity of the models compared with the models derived using the non-conservative bootstrap variance estimator. The non-conservative estimates of the variance, and therefore the instability, were underestimated for more complex models, and the models that were non-conservatively selected and assessed consequently had optimistic estimates of their predictive capacity relative to the *conservative* estimates. The less complex models selected using the *conservative* variance estimator either had smaller confidence intervals for the per sample prediction probabilities (Controls vs. NDB) or fewer samples with wide confidence intervals (NDB vs. HGD), which indicates that the less complex models are more stable, and that they may have increased generalisability to new samples, i.e., they are potentially more robust [23].

However, we observed evidence of residual model instability in the least complex classification model for NDB vs. HGD, which had considerable variance in the prediction probabilities for 7 out of the 38 samples. This contrasts with the least complex model for Controls vs. NDB where all of the samples had small to moderate sized confidence intervals which did not affect the worst case estimates of specificity and sensitivity derived from the confidence interval limits. This suggests that the NDB vs. HGD biomarker discovery process may benefit from the inclusion of more samples as it has been observed that the variance of prediction errors decreases with larger sample sizes [14]. We are undertaking additional blood collections to increase the number of NDB and HGD samples for our biomarker development studies.

When applied to our miRNA sequencing data, StaVarSel selected *miR-324-5p*/let-7b-5p and *miR-17-5p*/miR-126-5p as the top two frequently selected miRNA-ratios from NDB vs. HGD. Our previously reported 5 miRNA-ratio predictive model that was developed to differentiate between Controls combined with NDB vs. oesophageal adenocarcinoma, using a high throughput qPCR based technology [6], also contained *miR-324-5p* and *miR-17-5p*. Other researchers have identified circulating sEV *miR-324-5p* and *miR-17-5p* as potential biomarkers in gastric cancer [24,25].

The biomarker models derived in this study have the potential, if validated and assessed as potentially being cost effective, to be readily translated into use with standard techniques in pathology laboratories. The methods developed in this study for increasing the robustness of stable variable selection within cross validation have the potential to improve progress from biomarker discovery to biomarker driven translational research.

## 4. Materials and Methods

### 4.1. Patient Recruitment and Sample Selection

Patients were recruited as previously described [6]. Recruitment took place at Flinders Medical Centre (South Australia, Australia) and at Westmead Hospital (New South Wales, Australia). All patients had blood collected prior to endoscopy and serum was processed as previously described [26].

Samples from the following groups were selected for this study:

Controls (n = 14, all male). This group of patients had a visibly normal oesophageal mucosa at endoscopy. The median age was 64.7 years (min 21.2, max 75.6).

Non-dysplastic Barrett’s oesophagus (NDB; n = 21, all male). This group of patients had endoscopically visible Barrett’s oesophagus, which was confirmed by histology to be negative for dysplasia. The median age was 61.9 years (min 43.9, max 79.2). The median Prague C length was 3.0 cm (min 0, max 14.0), and the median Prague M length was 5.0 cm (min 2.0, max 15.0).

Barrett’s oesophagus with high grade dysplasia (HGD; n = 19, all male). This group of patients had a histological confirmation of high-grade dysplasia within the Barrett’s oesophagus segment. In one patient, the presence of T1a cancer was also noted on the histology report. The median age was 58.0 years (min 42.3, max 83.4). The median Prague C length was 3.0 cm (min 0.0, max 11.0), and the median Prague M length was 5.0 cm (min 1.0, max 13.0).

### 4.2. Extracellular Vesicle Isolation

As described previously [27], for small extracellular vesicle isolation, 1 mL aliquots of serum were retrieved from −80 degrees Celsius storage, quickly thawed, and centrifuged at 16,000× *g* at 4 °C for 30 min to exclude larger microparticles. Subsequently, 250 μL supernatant from each sample was processed with an ExoQuick^TM^ kit (System Biosciences, Palo Alto, CA, USA; EXOQ20A-1) according to the manufacturer’s protocol. Samples were incubated with ExoQuick^TM^ at 4 °C for 16 h. The pellet isolated from each sample was resuspended with 50 μL phosphate buffered saline (PBS). We have previously confirmed that pellets obtained from serum using ExoQuick^TM^ contain particles consistent in size with exosomes (30–150 nm), using a Nanosight LM10 Nanoparticle Analysis System and Nanoparticle Tracking Analysis Software (Nanosight Ltd., Malvern, UK). We refer to these as small EVs, as recommended in the Minimal Information for Studies of Extracellular Vesicles 2018 Guidelines [28].

### 4.3. RNA Extraction from Serum Small Extracellular Vesicles

As described previously [27], extraction of miRNA from small EVs was performed using the commercial miRNeasy Serum/Plasma kit (QIAGEN, #217184, Dusseldorf, Germany) according to the manufacturer’s protocol. Five microlitre (0.1 picomole) of each of the synthetic RNA molecules ath-miR-159a and cel-miR-54 (Shanghai Genepharma Co., Ltd., Shanghai, China) were added to the 500 L QIAzol vesicle lysate before further processing. Twenty-four microlitres of RNase-free ultrapure water was used for the final RNA elution step.

### 4.4. Qiagen Next Generation Sequencing of miRNA

Serum small EV miRNAs from patients were profiled using NGS by Qiagen (Hilden, Germany). As described previously [27], the library preparation was done using the QIAseq miRNA Library Kit (Qiagen, Hilden, Germany). A total of 5 μL total RNA was converted into miRNA NGS libraries. Adapters containing unique molecular identifiers were ligated to the RNA. The RNA was then converted to cDNA, and the cDNA was amplified, with the addition of indices, using 22 cycles of PCR. The samples were then purified, and library preparation QC was performed. One of the HGD samples failed library QC and was not sequenced. The libraries were pooled in equimolar ratios based on the quality of the inserts and the concentration measurements. The library pools were quantified using qPCR and were then sequenced on a NextSeq550 sequencing instrument according to the manufacturer instructions, using a single-end protocol for 75 bp with an average of 12 million reads per sample.

Raw data was de-multiplexed and FASTQ files for each sample were generated using the bcl2fastq software (Illumina Inc., San Diego, CA, USA). All primary analysis was carried out using CLC Genomics Server 20.0.4. The workflow “QIAseq miRNA Quantification” of CLC Genomics Server with standard parameters was used to map the reads to miRBase version 22. In short, the reads were processed by (1) trimming of the common sequence, UMI and adapters, and (2) filtering of reads with length < 15 nt or length > 55 nt. They were then deduplicated using their Unique Molecular Identifier (UMI). Reads were grouped into UMI groups when they (1) started at the same position based on the end of the read to which the UMI is ligated (i.e., Read2 for paired data), (2) were from the same strand, and (3) had identical UMIs. Groups that contained only one read (singletons) were merged into non-singleton groups if the singleton’s UMI could be converted to a UMI of a non-singleton group by introducing a SNP (the biggest group was chosen). All reads that did not map to miRBase with perfect matches were mapped to the human genome GRCh38 with ENSEMBL GRCh38 version 98 annotation. This was carried out using the “RNA-Seq Analysis” workflow of CLC Genomics Server with standard parameters. For normalisation, the trimmed mean of M-values method based on log-fold and absolute gene-wise changes in expression levels between samples (TMM normalisation) was used [29]. The resultant normalised levels were expressed in counts per million (CPM).

### 4.5. Prefiltering of miRNAs and Samples

miRNAs with average CPM < 5 were removed and miRNAs without sequencing counts in 100% of samples were also removed prior to data analysis. For the assessment of the samples, the CPM data for each miRNA was centered to a mean = 1, and samples with mean CPM across the miRNAs greater than the overall mean +/− 3SD were identified. This approach identified two samples that were subsequently determined to be haemolysed according to the molecular analysis in the next Section 4.6.

### 4.6. Molecular Detection of Haemolysis

The levels of miR-451a, miR-16-5p, and miR-486-5p are enriched in red blood cells and the levels of these miRNAs in cell free preparations are proportional to the degree of haemolysis [30,31]. We, therefore, assessed the relative levels of these miRNAs in all samples in addition to visually inspecting the serums. Data for each miRNA were centered so that the average was one across the samples. Samples were then classified as haemolysed when the average of the levels of miR-451a, miR-16-5p, and miR-486-5p exceeded the average of all samples by two standard deviations. One of the control samples and one of the HGD samples were classified as haemolysed and were not used for analysis.

### 4.7. miRNA Biomarkers Selection

Analyses were performed using R, version 3.4.3. The use of gene expression ratios provides good sensitivity and specificity in RNA biomarker studies [6,12]. We consequently calculated the ratio of the level of each miRNA with every other miRNA [12].

The miRNA-ratios were subjected to nested 2-stage cross validation consisting of an outer loop and inner loops over the training sets. Prefiltering was done within each training set; miRNA ratios with high variation in both of the comparison groups were removed (coefficient of variation > 300%), and the miRNA ratios were then pre-filtered (Mann–Whitney U-test at *p* > 0.2) to remove non-informative ratios [32].

Lasso regression (least absolute shrinkage and selection operator) was used to select miRNA-ratios which, when combined in a multivariable logistic regression model, had predictive capacity to differentiate Control patients from patients with non-dysplastic Barrett’s oesophagus (NDB), and to differentiate patients with NDB from patients with high grade dysplasia (HGD).

### 4.8. Standard Nested Cross Validation (2-Stage)

As described previously [12], we utilised leave-one-out cross validation (LOOCV) in the outer loop of a standard nested cross validation to generate held-out test samples that would not be used in optimisation and variable selection, and then utilised repeated (100× in an inner loop) 10-fold cross validation within each training set (using the cv.glmnet function from the R glmnet package v4.1-4) to optimise the penalisation parameter lambda for Lasso regression. The average of the lambda estimates from all of the training sets was then used in Lasso regression in each of the outer loop leave-one-out cross validation training sets to predict the corresponding held-out test samples. The generalised prediction error was calculated at a prediction probability cut-point threshold of 0.5.

In addition to optimising the Lasso regression model regularisation at the level that produced the minimum cross validated prediction error (lambda.min), we repeated the modelling using more stringent regularisation, based on the one standard error rule, to reduce model complexity [23].

### 4.9. Stabilised Nested Cross Validation (StaVarSel, 3-Stage)

In the standard nested cross validation scheme, when applied to molecular data, Lasso regression typically fits a different optimised multi-variable model to each training set. Each of these models may contain miRNA-ratios that are different to those selected in other training sets, and some of the miR-ratios may be selected infrequently. This can result in an unstable cross validation. To address this issue, Rosenburg et al. (2010) [33], in an approach that was motivated by the stability method proposed by Meinshausen and Bühlmann (2010) [8], identified variables that were selected in at least 50% of the cross validation training sets in order to build a final set of stable variables. As described previously [12], we extended this method by using an extra round of cross validations over the training sets to determine the optimum number of miRNA-ratios, from a ranked list of the most frequently selected miRNA-ratios, that produced the lowest average prediction error [12]. This is an iterative step-forward process. At each step, an increasing number of the frequently selected miR-ratios are used in an LOOCV within each training set to build multivariable logistic regression models and derive a prediction error. The set of miRNA-ratios that produced the lowest average prediction error across all of the training sets are then used to build a final model in each training set using generalised linear logistic regression, i.e., the equivalent to using the Relaxed Lasso of Meinshausen (2007) with 𝜙 = 0 [34]. Each training set model is then used to generate a prediction probability for the corresponding held-out test sample that was excluded from the model selection process in the outer loop. Estimates of the generalisable specificity and sensitivity of the final set of miRNA-ratios are derived from the set of prediction probabilities for the held out samples. In our initial implementation we derived confidence intervals, using 2000 bootstrap samples from the prediction probabilities of the held out samples, for the sensitivities and specificities at each threshold level in a Receiver Operating Characteristic (ROC) curve analysis [12,35].

### 4.10. Conservative Subsampling Based Estimation of Confidence Intervals Based on a Hold out Test

We used leave-one-out cross validation (LOOCV) in the outer loop of our previously implemented stablised nested cross validation (StaVarSel) to make the stable biomarker selection computationally tractable. However, the use of LOOCV in the outer loop of a standard nested cross validation has conceptually limited the range of methods available for estimating the variance of prediction errors to either a standard naive biased estimator that assumes that the prediction probabilities are normally distributed, or a non-parametric resampling method (e.g., bootstrap) that is applied simultaneously over the prediction probabilities (one per sample) of the held out samples in the outer loop. We previously utilised a non-parametric bootstrap approach for estimation of the variance of prediction errors. However, no unbiased estimator of the variance of prediction errors exists for cross validation [13], and these standard methods can result in a large underestimate of the variance (i.e., they are anti-conservative) [14]. It has therefore been suggested that the only robust and practically useful approach is to derive *conservative* estimates, i.e., that do not underestimate the variance, by using holdout tests [14,22].

Furthermore, the non-conservative bootstrap estimator cannot be used to estimate the prediction error variance when there is perfect separation between the groups. We therefore investigated alternative approaches to this problem and determined that the estimation of the generalisable predictive capacity of biomarkers does not need to be restricted to the standard LOOCV framework. A training set can be resampled repeatedly to produce a range of prediction probabilities for a held out sample from which non-parametric 95% confidence intervals can be derived. This approach can also be used with k-fold cross validation and is therefore applicable to the inner loops in each training set in a nested cross validation, and therefore applicable to the model selection process.

Although this resampling approach results in each resample derived estimate being biased relative to a single estimate derived from the entire training set, what is gained is the ability to estimate the variance *conservatively* for each sample. The Jackknife resampler is used for this instead of the bootstrap because the limiting distribution is non-normal, and because the Jackknife estimator is never downwardly biased [18,36], i.e., it is *conservative*.

We therefore estimated the prediction error variance by repeated (1000×) n-2 subsampling without replacement (i.e., monte carlo n-2 subsampling, which is also referred to as a general Jackknife). The results of these analyses are presented in Prediction Probability plots in which the average prediction probability (y-axis) for each sample (x-axis) is plotted as a circle, and the error bars represent the 95% confidence interval of the prediction probabilities per sample. A similar approach was used within the inner loops of the StaVarSel stabilised nested cross validation to derive the optimum number of biomarkers from the ranked list of frequently selected biomarkers.

A schema of the nested cross validation with both the stable variable selection and the *conservative* prediction error variance estimation is shown below in Figure 1.

## Data Availability

microRNA data are available at https://www.ncbi.nlm.nih.gov/geo/query/acc.cgi?acc=GSE227710 (accessed on 9 March 2023). The NCBI GEO accession number is GSE227710. R code is freely available from GitHub upon request to the corresponding authors. Please cite this paper if you use this R-code. All analyses were performed using R software version 4.2.0 Copyright (C) 2022 The R Foundation for Statistical Computing Platform: aarch64-apple-darwin20 (64-bit). Microsoft Excel version 16.70 was used for the generation of Appendix A.

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
