# Peer review of "A Method for Increasing the Robustness of Stable Feature Selection for Biomarker Discovery in Molecular Medicine Developed Using Serum Small Extracellular Vesicle Associated miRNAs and the Barrett’s Oesophagus Disease Spectrum"

_ijms, 2023, doi:10.3390/ijms24087068_

Round 1

Reviewer 1 Report

Summary and Comments:

This paper introduced the development of a new computational method in the application of biomarker selection. This new method was built on top of a previous method developed by the same group called StaVarSel, which dramatically improved the estimated generalisable predictive capacity of serum miRNA biomarkers in disease detection and prediction. The new method provided added benefit of decreasing the model complexity and increased stability.

Overall, this paper is well written and could be of interest to the general audience in biomarker selection and computational method development fields. Please see detailed comments below.

Detailed Comments:

1.     In Figures, for the ease of read, it might be better to refer each panel in the caption as a. b. c. d. as the actual figures and body texts show, instead of using Right Panel, Left Panel, etc.

2.     Page 5 line 207-211, it would be helpful if the authors could explain the rationality behind the miRNA and ratio selection.

3.     Figure S1 is empty?

Reviewer 2 Report

The manuscript by Mayne et al details a method for biomarker discovery for Barrett’s Oesophagus Disease by reducing the instability in biomarker selection. The authors improved their previously proposed stable variable selection method by combining it with a conservative prediction error variance estimator. This approach improved the predictive capacity of the biomarker models. The paper is very well-written and very detailed. My only suggestion would be to tabulate the differences/results of both methods.  It will provide a better understanding of the differences and improvements of the second model form the first one.

Reviewer 3 Report

Very interesting work of validating the new algorithm incorporating more precise feature selection and prediction with a case study of oesophageal disease biomarker discovery. The manuscript is well-written and the methodology is rigorous and adequate. Two comments for the authors’ consideration to further improve this submission:

#1. Although the authors made clear references to a previous manuscript (i.e., citation 28) regarding some materials and methods (i.e., “as described previously…”), the reviewer still feels it would be necessary to have those pieces of information as a part of the supplementary materials of this manuscript for the readers’ convenience.

#2. The reviewer was also wondering whether ridge or elastic net were considered and evaluated in comparison to Lasso used in the predictive model. 

Additional minor comments: 

#1. Line 54 - please confirm the grammar/punctuation of this sentence 

#2. Line 404 - please confirm the next section “4.6” vs “5.6”. 

#3. Please confirm the consistency of the capitalization of subtitles

#4. Please confirm the consistency of spells vs symbols of numbers throughout the manuscript when applicable
